# Doubly-Asynchronous Value Iteration:
# Making Value Iteration Asynchronous in Actions

**Tian Tian    Kenny Young    Richard S. Sutton**
University of Alberta and Alberta Machine Intelligence Institute
Edmonton, Alberta, Canada
{ttian, kjyoung, rsutton}@ualberta.ca

## Abstract

Value iteration (VI) is a foundational dynamic programming method, important for learning and planning in optimal control and reinforcement learning. VI proceeds in batches, where the update to the value of each state must be completed before the next batch of updates can begin. Completing a single batch is prohibitively expensive if the state space is large, rendering VI impractical for many applications. Asynchronous VI helps to address the large state space problem by updating one state at a time, in-place and in an arbitrary order. However, Asynchronous VI still requires a maximization over the entire action space, making it impractical for domains with large action space. To address this issue, we propose *doubly-asynchronous value iteration* (DAVI), a new algorithm that generalizes the idea of asynchrony from states to states and actions. More concretely, DAVI maximizes over a sampled subset of actions that can be of any user-defined size. This simple approach of using sampling to reduce computation maintains similarly appealing theoretical properties to VI without the need to wait for a full sweep through the entire action space in each update. In this paper, we show DAVI converges to the optimal value function with probability one, converges at a near-geometric rate with probability $1 - \delta$, and returns a near-optimal policy in computation time that nearly matches a previously established bound for VI. We also empirically demonstrate DAVI's effectiveness in several experiments.

## 1   Introduction

Dynamic programming has been used to solve many important real-world problems, including but not limited to wireless networks (Levorato et al., 2012; Liu et al., 2017, 2019) resource allocation (Powell et al., 2002), and inventory problems (Bensoussan, 2011). Value iteration (VI) is a foundational dynamic programming algorithm central to learning and planning in optimal control and reinforcement learning.

VI is one of the most widely studied dynamic programming algorithm (Williams and Baird III, 1993; Bertsekas and Tsitsiklis, 1996; Puterman, 1994). VI starts from an arbitrary value function and proceeds by updating the value of all states in batches. The value estimate for each state in the state space $\mathcal{S}$ must be computed before the next batch of updates will even begin:

$$v(s) \leftarrow \max_{a \in \mathcal{A}} \left\{ r(s, a) + \gamma \sum_{s' \in \mathcal{S}} p(s'|s, a)\bar{v}(s') \right\} \quad \text{for all } s \in \mathcal{S}, \tag{1}$$

where $s' \in \mathcal{S}$ is the next state, $p(s'|s, a)$ is the transition probability, and $r(s, a)$ is the reward. VI maintains two arrays of real values $v, \bar{v}$, both the size of state space $S \doteq |\mathcal{S}|$, with $\bar{v}$ being used to make the update and $v$ being used to keep track of the updated values. In domains with large state

space, for example wireless networks where the state-space of the network scales exponentially in the number of nodes, computing a single batch of state value updates using VI is prohibitively expensive, rendering VI impractical.

An alternative to updating the state values in batches is to update the state values one at a time, using the most recent state value estimates in the computation. Asynchronous value iteration (Williams and Baird III, 1993; Bertsekas and Tsitsiklis, 1996; Puterman, 1994) starts from an arbitrary value function and proceeds with in-place updates:

$$v(s_n) \leftarrow \max_{a \in \mathcal{A}} \left\{ r(s_n, a) + \gamma \sum_{s' \in \mathcal{S}} p(s'|s_n, a) v(s') \right\}, \tag{2}$$

where $s_n \in \mathcal{S}$ is the sampled state for update in iteration $n$ with all other states $s \neq s_n$ value remain the same.

Although Asynchronous VI helps to address domains with large state space, it still requires a sweep over the actions for every update due to the maximization operation over the look-ahead values: $L^v(s, a) \doteq r(s, a) + \gamma \sum_{s' \in \mathcal{S}} p(s'|s, a) v(s')$ for a given $v \in \mathbb{R}^S$. Evaluating the look-ahead values, as large as the action space, can be prohibitively expensive in domains with large action space. For example, running Asynchronous VI would be impractical in fleet management, where the action set is exponential in the size of the fleet.

There have been numerous works that address the large action space problem. Szepesvári and Littman (1996) proposed an algorithm called sampled-max, which performs a max operation over a smaller subset of look-ahead values. Their algorithm resembles Q-learning, requiring a step-size parameter that needs additional tuning. However, their algorithm does not converge to the optimal value function $v^*$. Williams and Baird III (1993) presented convergence analysis on a class of asynchronous algorithms, including some that may help address the large action space problem. Hubert et al. (2021) and Danihelka et al. (2022) explore ways to achieve policy improvement while sampling only a subset of actions in Monte Carlo Tree Search. Ahmad et al. (2020) focused on the generation of a smaller candidate action set to be used in planning with continuous action space.

We consider the setting where we have access to the underlying model of the environment and propose a variant of Asynchronous VI called *doubly-asynchronous value iteration* (DAVI) that generalizes the idea of asynchrony from states to states and actions. Like Asynchronous VI, DAVI samples a state for the update, eliminating the need to wait for all states to complete their update in batches. Unlike Asynchronous VI, DAVI samples a subset of actions of any user-defined size via a predefined sampling strategy. It then computes only the look-ahead values of the corresponding subset and a best-so-far action, and updates the state value to be the maximum over the computed values. The intuition behind DAVI is the idea of incremental maximization, where maximizing over a few actions could improve the value estimate for a certain state, which helps to evaluate other state-action pairs in subsequent back-ups. This simple approach of using sampling to reduce computation maintains similarly appealing theoretical properties to VI. In particular, we show DAVI converges to $v^*$ with probability 1 and at a near-geometric rate with probability $1 - \delta$. Additionally, DAVI returns an $\epsilon$-optimal policy with probability $1 - \delta$ using

$$O \left( m S H_{\gamma, \epsilon} \left( \ln \left( \frac{S H_{\gamma, \epsilon}}{\delta} \right) \Big/ \ln \left( \frac{1}{1 - q_{min}} \right) \right) \right) \tag{3}$$

elementary operations, where $m$ is the size of the action subset, $H_{\gamma, \epsilon}$ is a horizon term, and $q_{min}$ is the minimum probability that any state-action pair is sampled. We also provide a computational complexity bound for Asynchronous VI, which to the best of our knowledge, has not been previously reported. Our computational complexity bounds for both DAVI and Asynchronous VI nearly match a previously established bound for VI (Littman et al., 1995). Finally, we demonstrate DAVI's effectiveness in several experiments.

Related work by Zeng et al. (2020) uses an incremental maximization mechanism that is similar to ours. However, their work focuses on a different setting from ours, an asynchronous parallel setting, where an algorithm is hosted on several different machines running concurrently without waiting for synchronization of the computations. Aside from this difference, their work considers the case where the agent has access to only a generative model (Kearns et al., 2002) of the environment, whereas we assume full access to the transition dynamics and reward function. They also provided a computational complexity bound, which differs significantly from ours due to differences in settings.

## 2 Background

We consider a discounted Markov Decision Process $(\mathcal{S}, \mathcal{A}, r, p, \gamma)$ consisting of a finite set of states $\mathcal{S}$, a finite set of actions $\mathcal{A}$, a stationary reward function $r : \mathcal{S} \times \mathcal{A} \to [0, 1]$, a stationary transition function $p : \mathcal{S} \times \mathcal{A} \times \mathcal{S} \to [0, 1]$, and a discount parameter $\gamma \in [0, 1)$. The stochastic transition between state $s \in \mathcal{S}$ and the next state $s' \in \mathcal{S}$ is the result of choosing an action $a \in \mathcal{A}$ in $s$. For a particular $s, a \in \mathcal{S} \times \mathcal{A}$, the probability of landing in various $s'$ is characterized by a transition probability, which we denote as $p(s'|s, a)$.

For this paper, we use $\Pi$ to denote a set of deterministic Markov polices $\{\pi : \mathcal{S} \to \mathcal{A}\}$. The value function of a state evaluated according to $\pi$ is defined as $v_\pi(s) \doteq \mathbb{E}\left[\sum_{n=0}^{\infty} \gamma^n r(s_n, a_n)|s_0 = s\right]$ for a $s \in \mathcal{S}$. The optimal value function for a state $s \in \mathcal{S}$ is then $v^*(s) \doteq \max_{\pi \in \Pi} v_\pi(s)$ and there exists a deterministic optimal policy $\pi^*$ for which the value function is $v^*$. For the rest of the paper, we will consider $v$ as a vector of values of $\mathbb{R}^S$. For a fixed $\epsilon > 0$, a policy $\pi$ is said to be $\epsilon$-optimal if $v_\pi \geq v^* - \epsilon \mathbb{1}$. Finally, we will use $\|\cdot\|$ to denote infinity norm (i.e, $\|v\| = \max_i |v_i|$) and $A$ to denote the size of the action space.

## 3 Making value iteration asynchronous in actions

We assume access to the reward and transition probabilities of the environment. In each iteration $n$, DAVI samples a state $s_n$ for the update and $m$ actions from the action set $\mathcal{A}$. It then computes the corresponding look-ahead values of the sampled actions and the look-ahead value of a best-so-far action $\pi_n(s_n)$. Finally, DAVI updates the state value to the maximum over the computed look-ahead values. The size of the action subset can be any user-defined value between $1$ and $A$. To maintain a best-so-far action, one for every state amounts to maintaining a deterministic policy in every iteration. Recall $L^v(s, a) \doteq r(s, a) + \gamma \sum_{s' \in \mathcal{S}} p(s'|s, a) v(s')$ for a given $s, a \in \mathcal{S} \times \mathcal{A}$ and $v \in \mathbb{R}^S$, the pseudo-code for DAVI is shown in Algorithm 1.

---

**Algorithm 1:** DAVI$(m, p, q, \tau)$

**Input:** State sampling distribution $p \in \Delta(S)$
**Input:** A potentially state conditional distribution over the sets of actions of size $m$ denoted by $q$
**Input:** Number of iterations $\tau$, see Corollary 1 for how to choose $\tau$ to obtain an $\epsilon$-optimal policy with high probability
1   Initialize the value function $v_0 \in \mathbb{R}^S$
2   Initialize the policy $\pi_0$ to an arbitrary deterministic policy
3   **for** $n = 0, \cdots, \tau$ **do**
4      Sample a state from $p$
5      Sample $m$ actions from $q(\cdot|s_n)$ and let it be $\mathcal{A}_n$
6      Let $a_n^* = \arg\max_{a \in A_n} L^{v_n}(s, a)$ with ties broken randomly
7      Value back-up step:
8      $v_{n+1}(s) = \begin{cases} \max\left\{L^{v_n}(s, a_n^*), L^{v_n}(s, \pi_n(s))\right\} & \text{if } s = s_n \\ v_n(s) & \text{otherwise} \end{cases}$
9      Policy improvement step:
10      $\pi_{n+1}(s) = \begin{cases} a_n^* & \text{if } s = s_n \text{ and } L^{v_n}(s, a_n^*) > L^{v_n}(s, \pi_n(s)) \\ \pi_n(s) & \text{otherwise} \end{cases}$
11   **end**
12   **return** $v_n, \pi_n$

---

Note that the policy will only change if there is another action in the newly sampled subset whose look-ahead value is strictly better than the current best-so-far action's look-ahead value.

## 4 Convergence

We show in Theorem 1 that DAVI converges to the optimal value function despite only maximizing over a subset of actions in each update. Before showing the proof, we establish some necessary definitions, lemmas, and assumptions.

**Definition 1 ($q_{min}$ and $p_{min}$)** *Recall $p$ is a distribution over states and $q$ is a potentially state conditional distribution over the sets of actions of size $m$. Then, we will use $\tilde{q}(s,a)$ to denote the joint probability that a single state $s \in \mathcal{S}$ is sampled for update with a particular action $a$ included in the set sampled by $q$. Furthermore, let $q_{min} \doteq \min_{s,a} \tilde{q}(s,a)$ and $p_{min} \doteq \min_s p(s)$.*

**Definition 2 (Bellman optimality operator and policy evaluation)** *Let $T : \mathbb{R}^S \to \mathbb{R}^S$. For all $s \in \mathcal{S}$, define $Tv(s) \doteq \max_{a \in \mathcal{A}} L^v(s,a)$. Let $T_\pi : \mathbb{R}^S \to \mathbb{R}^S$. For a given $\pi$, for all $s \in \mathcal{S}$, define $T_\pi v(s) \doteq L^v(s,\pi(s))$.*

**Definition 3 (DAVI back-up operator $T_n$)** *Let $T_n : \mathbb{R}^S \to \mathbb{R}^S$. For a given $\mathcal{A}_n \sim \tilde{q}$, $\pi_n \in \Pi$, $s_n \in \mathcal{S}$, and for all $s \in \mathcal{S}$ and $v \in \mathbb{R}^S$, define*

$$T_n v(s) \doteq \begin{cases} \max_{a \in \mathcal{A}_n \cup \{\pi_n(s)\}} L^v(s,a) & \text{for } s = s_n \\ v(s) & \text{otherwise.} \end{cases} \tag{4}$$

We show in Appendix A that $T_n$ is a monotone operator.

**Assumption 1 (Initialization)** *We consider the following initialisations, (i) $v_0 = 0\mathbb{1}$, or (ii) $v_0 = -c\mathbb{1}$ for $c > 0$, or (iii) $v_0(s) \leq L^{v_0}(s,\pi_0(s))$ for all $s \in \mathcal{S}$.*

**Lemma 1 (Monotonicity)** *If DAVI is initialized according to (i),(ii), or (iii) of Assumption 1, the value iterates of DAVI, $(v_n)_{n \geq 0}$ is a monotonically increasing sequence: $v_n \leq v_{n+1}$ for all $n \in \mathbb{N}_0$, if $r(s,a) \in [0,1]$ for any $s,a \in \mathcal{S} \times \mathcal{A}$.*

Proof: See Appendix A.    ∎

**Lemma 2 (Boundedness (Williams and Baird III, 1993))** *Let $v_{\max} = \max_s v_0(s)$ and $v_{\min} = \min_s v_0(s)$ and recall that any reward $\in [0,1]$. If we start with any $(v_0, \pi_0)$, then applying DAVI's operation on the $(v_0, \pi_0)$ thereafter, will satisfy: $\min\{0, v_{\min}\} \leq V_n(s) \leq \max\left\{\frac{1}{1-\gamma}, v_{\max}\right\}$, for all $s \in \mathcal{S}$ and for all $n \in \mathbb{N}_0$.*

**Lemma 3 (Fixed-point iteration (Szepesvári, 2010))** *Given any $v \in \mathbb{R}^S$, and $T, T_\pi$ defined in Definition 2*

1. *$v_\pi = \lim_{n \to \infty} T_\pi^n v$ for a given policy $\pi$. In particular for any $n \geq 0$, $\|T_\pi^n v - v_\pi\| \leq \gamma^n \|v - v_\pi\|$ where $v_\pi$ is the unique function that satisfies $T_\pi v_\pi = v_\pi$.*

2. *$v^* = \lim_{n \to \infty} T^n v$ and in particular for any $n \geq 0$, $\|v^* - T^n v\| \leq \gamma^n \|v^* - v\|$, where $v^*$ is the unique function that satisfies $Tv^* = v^*$.*

**Theorem 1 (Convergence of DAVI)** *Assume that $\tilde{q}(s,a) > 0$ and $r(s,a) \in [0,1]$ for any $s,a \in \mathcal{S} \times \mathcal{A}$, then DAVI converges to the optimal value function with probability 1, if DAVI initializes according to (i),(ii), or (iii) of Assumption 1.*

Proof:

By the Monotonicity Lemma 1 and Boundedness Lemma 2, DAVI's value iterates $(v_n)_{n \geq 0}$ are a bounded and monotonically increasing sequence. By the monotone convergence theorem, $\lim_{n \to \infty} v_n = \sup_n v_n \doteq \bar{v}$. It remains to show that $\bar{v} = v^*$. We first show $\bar{v} \leq v^*$ and then show $\bar{v} \geq v^*$ to conclude that $\bar{v} = v^*$. We note that $v_n = T_{n-1} v_{n-1} \leq T v_{n-1}$, where $T$ is the Bellman optimality operator that satisfies the Fixed-point Lemma 3 (2). By the monotonicity of $T$ and the monontonicity of $v_n$'s, for any $n \geq 0$, $v_n \leq T^n v_0$. By taking the limit of $n \to \infty$ on both sides, we get $\bar{v} \leq v^*$.

Now, we show $\bar{v} \geq v^*$. Let $(n_k)_{k=0}^\infty$ be a sequence of increasing indices, where $n_0 = 0$, such that between $n_k$-th and $n_{k+1}$-th iteration, all state $s \in \mathcal{S}$ have been updated at least once with an action set containing $\pi^*(s)$. We note that the number of iterations between any $n_k$ and $n_{k+1}$ is finite with probability 1 since there is a finite number of states and actions, and all state-action pairs are sampled with non-zero probability. Then, for any state $s$, let $t(s,k)$ be an iteration index such that $s_{t(s,k)} = s$,

$n_k \leq t(s,k) \leq n_{k+1}$, and $\pi^*(s) \in \mathcal{A}_{t(s,k)}$, then

$$v_{n_{k+1}}(s) \geq v_{t(s,k)+1}(s) \qquad \text{by monotonicity of } v_n \tag{5}$$

$$= \max \left\{ \max_{a \in \mathcal{A}_{t(s,k)} \cup \{\pi_{t(s,k)}\} \backslash \{\pi^*(s)\}} L^{v_{t(s,k)}}(s,a), L^{v_{t(s,k)}}(s,\pi^*(s)) \right\} \geq L^{v_{t(s,k)}}(s,\pi^*(s)) \tag{6}$$

$$= T_{\pi^*} v_{t(s,k)}(s) \geq T_{\pi^*} v_{n_k}(s) \quad \text{by monotonicity of the operator.} \tag{7}$$

By the $n_{k+1}$-th iteration, $v_{n_{k+1}} \geq T_{\pi^*} v_{n_k}$, where $T_{\pi^*}$ is the policy evaluation operator that satisfies the Fixed-point Lemma 3 (1). Continuing with the same reasoning, $v_{n_k} \geq T_{\pi^*}^k v_0$ for any $k \geq 0$. By taking limit of $k \to \infty$ on both sides, we get $\bar{v} \geq v^*$. Altogether, $\bar{v} = v^*$. ∎

**Remark 1:** The initialization requirement in the Convergence of DAVI Theorem 1 can be relaxed to be any initialization, and DAVI will still converge to $v^*$ with probability 1. A more general proof can be found in Appendix A, which follows a similar argument to that of the proof for Theorem 4.3.2 of Williams and Baird III (1993). Intuitively, there exists a finite sequence of value back-up and policy improvement operations that will lead to one contraction, and if there are $l \in \mathbb{N}$ copies of such a sequence, this will lead to $l$ contractions. Once the value iterates contract into an "optimality-capture" region, where all the policies $\pi_n$ are optimal thereafter, DAVI is performing policy evaluations of an optimal policy. As long as all states are sampled infinitely often, the value iterates must converge to $v^*$. Finally, we show that such a finite sequence as a contiguous subsequence exists in an infinite sequence of operators generated by a stochastic process.

**Remark 2:** DAVI could be considered an Asynchronous Policy Iteration algorithm (Bertsekas and Tsitsiklis, 1996) since DAVI consists of a policy improvement step and a policy evaluation step. However, the algorithmic construct discussed by Bertsekas and Tsitsiklis (1996) does not exactly match that of DAVI with sampled action subsets. Consequently, we could not directly apply Proposition 2.5 of Bertsekas and Tsitsiklis (1996) to show DAVI's convergence. A more useful analysis is that of Williams and Baird III (1993); we could have applied their Theorem 4.2.6 to show DAVI's convergence after having shown that DAVI's value iterates are monotonically increasing in Lemma 1. However, Williams and Baird III (1993) provide no convergence rate or computational complexity. Therefore, we chose to present a different convergence proof, more closely related to the convergence rate proof in the next section.

## 5 Convergence rate

DAVI relies on sampling to avoid sweeps through the state and action space, which introduces additional error. Despite this, we show in Theorem 2 that DAVI converges at a near-geometric rate and nearly matches the computational complexity of VI.

**Theorem 2 (Convergence rate of DAVI)** *Assume $\tilde{q}(s,a) > 0$ and $r(s,a) \in [0,1]$ for any $s, a \in \mathcal{S} \times \mathcal{A}$, and also assume DAVI initialises according to (i), (ii), (iii) of Assumption 1. With $\gamma \in [0,1)$ and probability $1 - \delta$, the iterates of DAVI, $(v_n)_{n \geq 0}$ converges to $v^*$ at a near-geometric rate. In particular, with probability $1 - \delta$, for a given $l \in \mathbb{N}$,*

$$\|v^* - v_n\| \leq \gamma^l \|v^* - v_0\|, \tag{8}$$

*for any $n$ satisfying*

$$n \geq l \left\lceil \ln\left(\frac{Sl}{\delta}\right) \Big/ \ln\left(\frac{1}{1 - q_{min}}\right) \right\rceil, \tag{9}$$

*where $q_{min} = \min_{s,a} \tilde{q}(s,a)$.*

Proof: Recall from Lemma 1, we have shown $v_n \to v^*$ monotonically from below. From Theorem 1, we have also defined $(n_k)_{k=0}^{\infty}$ to be a sequence of increasing indices, where $n_0 = 0$, such that between the $n_k$-th and $n_{k+1}$-th iteration, all state $s \in \mathcal{S}$ have been updated at least once with an action set containing $\pi^*(s)$. At the $n_{k+1}$-th iteration, $v_{n_{k+1}} \geq T_{\pi^*} v_{n_k}$. This implies that at the $n_{k+1}$-th iteration, DAVI would have $\gamma$-contracted at least once:

$$0 \leq v^* - v_{n_{k+1}} \leq v^* - T_{\pi^*} v_{n_k}, \implies \|v^* - v_{n_{k+1}}\| \leq \|v^* - T_{\pi^*} v_{n_k}\|, \tag{10}$$

$$\|v^* - T_{\pi^*} v_{n_k}\| = \|T_{\pi^*} v^* - T_{\pi^*} v_{n_k}\| \leq \gamma \|v^* - v_{n_k}\| \tag{11}$$

$$\implies \|v^* - v_{n_{k+1}}\| \leq \gamma \|v^* - v_{n_k}\|. \tag{12}$$

Consider dividing $n \in \mathbb{N}$ iterations into uniform intervals of length $N$ such that the $i$-th interval is $(iN, (i+1)N-1)$. Let $\mathcal{E}_i(s)$ denote the event that at some iteration in the $i$-th interval, state $s$ has been updated with an action set containing $\pi^*(s)$. Therefore, an occurrence of event $\mathcal{E}_i \doteq \cap_{s \in \mathcal{S}} \mathcal{E}_i(s)$ would mean that at $(i+1)N$-th iteration, $v_{(i+1)N}$ would have contracted at least once. Then, on the event $\mathcal{E} \doteq \cap_i \mathcal{E}_i = \cap_i \cap_{s \in \mathcal{S}} \mathcal{E}_i(s)$, there have been at least $l$ $\gamma$-contraction after $n$ iterations.

We would like $\mathbb{P}(\mathcal{E}) \geq 1 - \delta$ or alternatively the probability of failure event $\mathbb{P}(\mathcal{E}^c) \leq \delta$, for some $\delta > 0$. However, just how large should $N$ be in order to maintain a failure probability of $\delta$? To answer this question, we first bound $\mathbb{P}(\mathcal{E}^c)$ using union bound:

$$\mathbb{P}(\mathcal{E}^c) = \mathbb{P}(\cup_i \cup_{s \in \mathcal{S}} \mathcal{E}_i^c(s)) \leq \sum_{i=1}^{l} \sum_{s \in \mathcal{S}} \mathbb{P}(\mathcal{E}_i^c). \tag{13}$$

From Definition 1, $\tilde{q}(s, \pi^*(s))$ is the joint probability that a single state $s$ is sampled for update with $\pi^*(s)$ included in the action subset sampled by $q$. Then, the probability that state $s$ is not updated with an action set containing $\pi^*(s)$ in $N$ iterations is $(1 - \tilde{q}(s, \pi^*(s)))^N$. Continuing from (13),

$$\mathbb{P}(\mathcal{E}^c) \leq \sum_{i=1}^{l} \sum_{s \in \mathcal{S}} (1 - \tilde{q}(s, \pi^*(s)))^N \leq Sl(1 - q_{min})^N, \tag{14}$$

where $q_{min} \doteq \min_{s,a} \tilde{q}(s, a)$. Now set $Sl(1 - q_{min})^N \leq \delta$ and solve for $N$,

$$N \geq \ln\left(\frac{\delta}{Sl}\right) \Big/ \ln(1 - q_{min}). \tag{15}$$

Thus, with probability at least $1 - \delta$, within

$$n = l \left\lceil \ln\left(\frac{Sl}{\delta}\right) \Big/ \ln\left(\frac{1}{1 - q_{min}}\right) \right\rceil \tag{16}$$

iterations DAVI will have $\gamma$-contracted at least $l$ times. ∎

**Remark:** We note that $p, q$ could be non-stationary and potentially chosen adaptively based on current value estimates, which is an interesting direction for future work.

**Corollary 1 (Computational complexity of obtaining an $\epsilon$-optimal policy)** *Fix an $\epsilon \in (0, \|v^* - v_0\|)$, and assume DAVI initialises according to (i), (ii), or (iii) of Assumption 1. Define*

$$H_{\gamma,\epsilon} \doteq \ln\left(\frac{\|v^* - v_0\|}{\epsilon}\right) \Big/ 1 - \gamma \tag{17}$$

*as a horizon term. Then, DAVI runs for at least*

$$\tau = H_{\gamma,\epsilon}\left(\ln\left(\frac{SH_{\gamma,\epsilon}}{\delta}\right) \Big/ \ln\left(\frac{1}{1 - q_{min}}\right)\right) \tag{18}$$

*iterations, returns an $\epsilon$-optimal policy $\pi_n : v_{\pi_n} \geq v^* - \epsilon\mathbb{1}$ with probability at least $1 - \delta$ using $O(mS\tau)$ elementary arithmetic and logical operations, where $m$ is the size of the action subset and $S$ is the size of the state space. Note that $\|v^* - v_0\|$ is unknown but it can be upper bounded by $\frac{1}{1-\gamma} + \|v_0\|$ given rewards are in $[0, 1]$.*

Proof: See Appendix A. ∎

**Remark:** As a straightforward consequence of Theorem 1 and Corollary 1, we show in Appendix A (Corollary 2) that DAVI returns an optimal policy $\pi^*$ with probability $1 - \delta$ within a number of computations that depends on the minimal value gap between the optimal action and the second-best action with respect to $v^*$.

We can compare the computational complexity bound for DAVI (the result of Corollary 1) to similar bounds for Asynchronous VI and VI. As far as we know, computational complexity bounds for Asynchronous VI have not been reported in the literature. We followed similar argument to Theorem 2 and Corollary 1 to obtain the computational complexity bound for Asynchronous VI in Appendix B.

Table 1: Computational complexity of VI, Asynchronous VI, DAVI

| Algorithms | Computational complexity | References |
|---|---|---|
| VI | $O\left(AS^2 H_{\gamma,\frac{\epsilon(1-\gamma)}{2\gamma}}\right)$ | Littman et al. (1995) |
| Asynchronous VI | $O\left(ASH_{\gamma,\epsilon}\frac{\ln\left(\frac{SH_{\gamma,\epsilon}}{\delta}\right)}{\ln\left(\frac{1}{1-p_{min}}\right)}\right)$ | This paper |
| DAVI | $O\left(mSH_{\gamma,\epsilon}\frac{\ln\left(\frac{SH_{\gamma,\epsilon}}{\delta}\right)}{\ln\left(\frac{1}{1-q_{min}}\right)}\right)$ | This paper |

Recall that $q_{min} \doteq \min_{s,a} \tilde{q}(s,a)$. Consider the case of uniform sampling of states and actions. Uniform sampling of the states results in a probability of $\frac{1}{S}$ of sampling a particular state, while uniform sampling of $m$ actions without replacement results in a probability of $\frac{m}{A}$ of including a particular action in the subset. Altogether $q_{min} = \frac{m}{SA}$. Uniform sampling of state and action subset is the best sampling strategy for the bound $O\left(mSH_{\gamma,\epsilon}\left(\ln\left(\frac{SH_{\gamma,\epsilon}}{\delta}\right)/\ln\left(\frac{1}{1-q_{min}}\right)\right)\right)$ because any non-uniform strategy would result in $\tilde{q}_{\min} < \frac{m}{SA}$. Suppose $\tilde{q}_{\min} = \frac{m}{SA}$, then $-m/\left(\ln(1-\frac{m}{SA})\right) \approx m/\left(\frac{m}{SA}\right) = SA$. Therefore, DAVI's computational complexity $O\left(mSH_{\gamma,\epsilon}\left(\ln\left(\frac{SH_{\gamma,\epsilon}}{\delta}\right)/\ln\left(\frac{1}{1-q_{min}}\right)\right)\right) \approx O\left(S^2 AH_{\gamma,\epsilon}\ln\left(\frac{SH_{\gamma,\epsilon}}{\delta}\right)\right)$. Likewise, uniform sampling of state will result in $p_{\min} = \frac{1}{S}$, and so it follows that $-1/\left(\ln\left(1-\frac{1}{S}\right)\right) \approx S$. Then, Asynchronous VI's computational complexity $O\left(ASH_{\gamma,\epsilon}\left(\ln\left(\frac{SH_{\gamma,\epsilon}}{\delta}\right)/\ln\left(\frac{1}{1-p_{min}}\right)\right)\right) \approx O\left(S^2 AH_{\gamma,\epsilon}\ln\left(\frac{SH_{\gamma,\epsilon}}{\delta}\right)\right)$.

Chen and Wang (2017) have established a lower bound on the computational complexity of planning in finite discounted MDP to be $\Omega(S^2 A)$. The importance of their result shows that no algorithm can escape this $S^2 A$ computational complexity. Both DAVI and Asynchronous VI computational complexity matches that of the lower-bound $\Omega(S^2 A)$ up to log terms in $S^2 A$ but have additional dependence on $H_{\gamma,\epsilon}$ and $\ln(1/\delta)$.

VI, Asynchronous VI, and DAVI all include a horizon term. The horizon term $H_{\gamma,\epsilon}$ improves upon the horizon term of $H_{\gamma,\frac{\epsilon(1-\gamma)}{2\gamma}} = \ln\left(\frac{2\gamma\|v^* - v_0\|}{\epsilon(1-\gamma)}\right)/(1-\gamma)$ that appears in the VI bound of Littman et al. (1995) when $\gamma > 0.5$. As VI does not require sampling, it has no failure probability. Thus, DAVI and Asynchronous VI both have an additional $\ln\left(\frac{1}{\delta}\right)$. We leave open the question of whether the additional log term $\ln(SH_{\gamma,\epsilon})$ in DAVI and Asynchronous VI is necessary.

The computational complexity of DAVI nearly matches that of VI, but DAVI does not need to sweep through the action space in every state update. Similar to Asynchronous VI, DAVI also does not need to wait for all states to complete their update in batches, as is the case of VI, making DAVI a more practical algorithm.

## 6 Experiments

DAVI relies on sampling to reduce the computation of each update, and the performance of DAVI can be affected by the sparsity of rewards. If a problem is like a needle in a haystack, where only one specific sequence of actions leads to a reward, then we do not anticipate uniform sampling to be beneficial in terms of total computation. An algorithm would still have to consider most states and actions to make progress in this case. On the other hand, we hypothesize that DAVI would converge faster than Asynchronous VI in domains with multiple optimal or near-optimal policies. To isolate the effect of reward sparsity from the MDP structure, we first test our hypothesis on several single-state MDP domains. However, solving a multi-state MDP is generally more challenging than solving a single-state MDP. In our second experiment, we examine the performance of DAVI on two sets of MDPs: an MDP with a tree structure and a random MPD.

The algorithms that will be compared in the experiments are VI, Asynchronous VI, and DAVI. We implement Asynchronous VI and DAVI using uniform sampling to obtain the states. DAVI samples a new set of actions via uniform sampling without replacement in each iteration.

## 6.1 Single-state experiment

This experiment consists of a single-state MDP with 10000 actions, all terminate immediately. We experiment on two domains: needle-in-the-haystack and multi-reward. Needle-in-the-haystack has one random action selected to have a reward of 1, with all other rewards set to 0. Multi-reward has 10 random actions with a reward of 1. The problems in this single-state experiment amount to brute-force search for the actions with the largest reward.

## 6.2 Multi-state experiment

This experiment consists of two sets of MDPs. The first set consists of a tree with a depth of 2. Each state has 50 actions, where each action leads to 2 other distinct next states. All actions terminate at the leaf states. Rewards are 0 everywhere except at a random leaf state-action pair, where reward is set to 1. With this construct, there are around 10000 states. The second set consists of a random MDP with 100 states, where each state has 1000 actions. Each action leads to 10 next states randomly selected from the 100 states with equal probability. All transitions have a 0.1 probability of terminating. A single state-action pair is randomly chosen to have a reward of 1. The $\gamma$ in all of the MDPs are 1.

## 6.3 Discussion

Figure 1 and Figure 2 show the performance of the algorithms. All graphs included error bars showing the standard error of the mean. Notice that all graphs started at 0 and eventually reached an asymptote unique to each problem setting. All graphs smoothly increased towards the asymptote except for Asynchronous VI in Figure 1 and VI in Figure 2, whose performances were step-functions [1]. The y-axis of each graph showed a state value averaged over 200 runs. The x-axes showed run-times, which have been adjusted for computations.

In Figure 1(a,b), DAVI with $m = 1$ was significantly different from that of DAVI with $m = 10, 100, 1000$, and DAVI with $m = 10, 100, 1000$ converged at a similar rate. While in Figure 2(a,b), all algorithms were significantly different. DAVI $m = 10$ in random MDP Figure 2(b) converged faster than any other algorithms. These results suggest that an ideal $m$ exists for each domain.

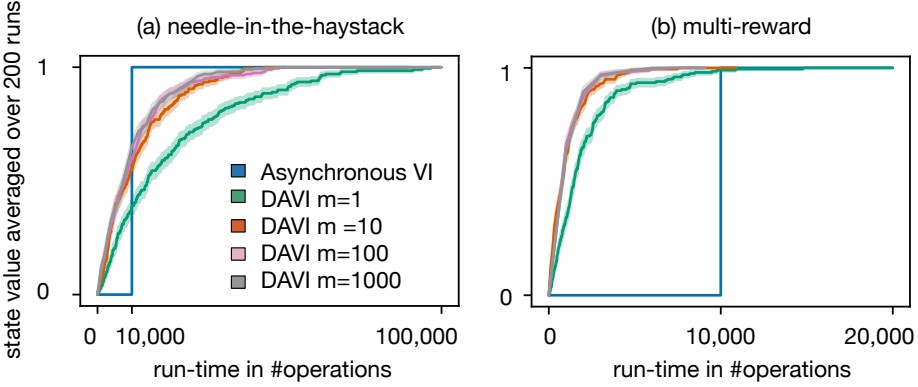

Figure 1: Single-state experiment with $10,000$ actions: (a) has only one action with a reward of 1 (b) has 10 actions with a reward of 1. The Asynchronous VI in this experiment is equivalent to VI since there is only one state. We run each instance 200 times with a new MDP generated each time. In each run, each algorithm is initialized to 0.

In the needle-in-the-haystack setting Figure 1(a), all four DAVI algorithms made some progress by the 10000 computation mark while Asynchronous VI stayed flat. Once Asynchronous VI finished computing the look-ahead value for all of the actions, it reached the asymptote immediately. On the other hand, DAVI might have been lucky in some of the runs, found the optimal action amongst the subset early on, logged it as a best-so-far action, and had a state value sustained at 1 thereafter. However, there could also be runs where sampling had the opposite effect.

---

[1]Asychronous VI in the single-state experiment is equivalent to VI since there is only one state.

Changing the reward structure by introducing a few redundant optimal actions into the action space increased the probability that an optimal action was included in the subset. In the multi-reward setting Fig. 1(b), DAVI with all settings of $m$ have essentially reached the asymptote by the 10000 mark. DAVI with all four action subset sizes reached the asymptote faster than Asynchronous VI. As expected, DAVI converged faster than Asynchronous VI in the case of multiple rewarding actions.

In the Figure 2(a), we saw a similar performance to that of the needle-in-the-haystack in the single-state experiment Figure 1(a). When we changed the MDP structure to allow for multiple possible paths that led to the special state with the hidden reward, as evident in Figure 2(b), DAVI with all settings of $m$ all reached the asymptote faster than Asynchronous VI and VI. As expected, DAVI converged faster than Asynchronous VI in the case of multiple near-optimal policies.

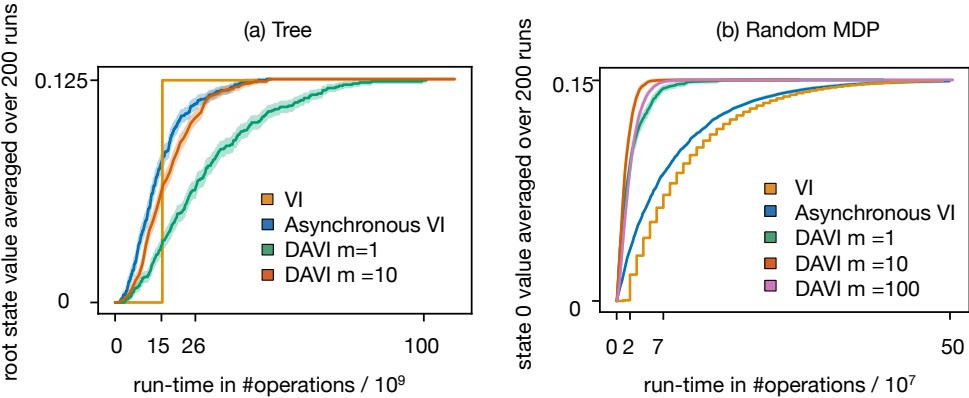

Figure 2: Multi-state-action needle-in-the-haystack experiment: (a) MDP with a tree structure (b) random MDP. We run each instance 200 times with a new MDP generated each time. In each run, all algorithms are initialized to 0.

We note that perhaps many other real-world problems resemble a setting like random-MDP more than needle-in-the-haystack, and hence the result on random-MDP may be more important. See Appendix C for additions experiments with rewards drawn from Normal and Pareto distributions.

# 7   Conclusion

The advantage of running asynchronous algorithms on domains with large state and action space have been made apparent in our studies. Asynchronous VI helps to address the large state space problem by making in-place updates, but it is still intractable in domains with large action space. DAVI is asynchronous in the state updates as in Asynchronous VI, but also asynchronous in the maximization over the actions. We show DAVI converges to the optimal value function with probability 1 and at a near-geometric rate with a probability of at least $1 - \delta$. Asynchronous VI also achieves a computational complexity closely matching that of VI. We give empirical evidence for DAVI's computational efficiency in several experiments with multiple reward settings.

Note that DAVI does not address the summation over the states: $\sum_{s' \in \mathcal{S}} p(s'|s, a)v(s')$ in the computation of the look-ahead values. If the state space is large, computing such a sum can also be prohibitively expensive. Prior works by Van Seijen and Sutton (2013) use "small back-ups" to address this problem. Instead of a summation of all successor states, they update each state's value with respect to one successor state in each update. Another possibility is to sample a subset of successor states to compute the look-ahead values at the cost of additional failure probability. Combining these techniques with DAVI is a potential direction for future work.

## Acknowledgments and Disclosure of Funding

We thank Tadashi Kozuno, Csaba Szepesvári, and Roshan Shariff for their valuable comments and feedback. The authors gratefully acknowledge funding from DeepMind, Amii, NSERC, and CIFAR.

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
