## Supplementary material

## A Auxilary proofs for DAVI's theoretical results

This section shows the proof of the supporting lemmas required in the proof of DAVI's convergence and convergence rate. We also include here a more general proof of the convergence of DAVI and each of the corollaries. The numbering of each lemma, corollary, and theorem corresponds to the main paper's numbering.

**Definition 4** *Recall $T_n : \mathbb{R}^S \to \mathbb{R}^S$. For a given $\mathcal{A}_n \sim \tilde{q}$, $\pi_n \in \Pi$, $s_n \in \mathcal{S}$, and for all $s \in \mathcal{S}$ and $v \in \mathbb{R}^S$,*

$$T_n v(s) \doteq \begin{cases} \max_{a \in \mathcal{A}_n \cup \{\pi_n(s)\}} L^v(s, a) & \text{if } s = s_n \\ v(s) & \text{otherwise}. \end{cases} \tag{19}$$

*Define $T_{\pi, s_n} : \mathbb{R}^S \to \mathbb{R}^S$. For a given $\pi \in \Pi$, $s_n \in \mathcal{S}$, and for all $s \in \mathcal{S}$ and $v \in \mathbb{R}^S$,*

$$T_{\pi, s_n} v(s) \doteq \begin{cases} L^v(s, \pi(s)) & \text{if } s = s_n \\ v(s) & \text{otherwise}. \end{cases} \tag{20}$$

*Then, the value iterates of DAVI evolves according to $v_{n+1} = T_n v_n$ for all $n \in \mathbb{N}_0$. Alternatively, $v_{n+1} = T_{\pi_{n+1}, s_n} v_n$ with $\pi_{n+1}(s)$ being the the action that satisfies $\max_{a \in \mathcal{A}_n \cup \{\pi_n(s)\}} L^{v_n}(s, a)$ for $s = s_n$ and $\pi_{n+1}(s) = \pi_n(s)$ for $s \neq s_n$. (i.e., $T_{\pi_{n+1}, s_n} v_n = T_n v_n$).*

**Definition 5 (Optimality capture region (Williams and Baird III, 1993))** *Define*

$$\Delta^v(s) = \min \left[ \left\{ \max_{a' \in \mathcal{A}} L^v(s, a') - L^v(s, a) \Big| a \in \mathcal{A} \right\} - \{0\} \right] \tag{21}$$

*as the difference between the look-ahead value with respect to $v$ of the greedy action and a second-best action for state $s$. Let $\Delta^{v^*} \doteq \min_{s \in \mathcal{S}} \Delta^{v^*}(s)$. Then, the optimality capture region is defined to be*

$$\left\{ v : \|v^* - v\| < \frac{\Delta^{v^*}}{2\gamma}, v \in \mathbb{R}^S \right\}. \tag{22}$$

**Lemma 4** *DAVI operators $T_n$ and $T_{\pi, s'}$ are monotone operators. That is given $v, u \in \mathbb{R}^S$ if $v \leq u$, then $T_n v \leq T_n u$ and $T_{\pi, s'} v \leq T_{\pi, s'} u$.*

Proof: Given any $v, u \in \mathbb{R}^S$ s.t. $v \leq u$, then

$$T_n v(s) = \begin{cases} \max_{a \in \mathcal{A}_n \cup \{\pi_n(s)\}} r(s, a) + \gamma \sum_{s'} p(s'|s, a) v(s') & \text{if } s = s_n \\ v(s) & \text{otherwise} \end{cases} \tag{23}$$

$$\leq \begin{cases} \max_{a \in \mathcal{A}_n \cup \{\pi_n(s)\}} r(s, a) + \gamma \sum_{s'} p(s'|s, a) u(s') & \text{if } s = s_n \\ u(s) & \text{otherwise} \end{cases} \tag{24}$$

$$= T_n u(s). \tag{25}$$

Given any $v, u \in \mathbb{R}^S$ s.t. $v \leq u$, then

$$T_{\pi, s_n} v(s) \doteq \begin{cases} r(s, \pi(s)) + \gamma \sum_{s'} p(s'|s, \pi(s)) v(s') & \text{for } s = s_n \\ v(s) & \text{otherwise} \end{cases} \tag{26}$$

$$\leq \begin{cases} r(s, \pi(s)) + \gamma \sum_{s'} p(s'|s, \pi(s)) u(s') & \text{for } s = s_n \\ u(s) & \text{otherwise} \end{cases} \tag{27}$$

$$= T_{\pi, s_n} u(s). \tag{28}$$

$\blacksquare$

**Lemma 1 (Monotonicity)** *The iterates of DAVI, $(v_n)_{n \geq 0}$ is a monotonically increasing sequence: $v_n \leq v_{n+1}$ for all $n \in \mathbb{N}_0$, if $r(s, a) \in [0, 1]$ for any $s, a \in \mathcal{S} \times \mathcal{A}$ and if DAVI is initialized according to (i),(ii), or (iii) of Assumption 1.*

Proof: We show $(v_n)_{n\geq 0}$ is a monotonically increasing sequence by induction. All inequalities between vectors henceforth are element-wise. Let $(s_0, s_1, ..., s_n, s_{n+1})$ be the sequence of states sampled for update from iteration 1 to $n+1$. By straight-forward calculation, we show $v_1 \geq v_0$. For all rewards in $[0, 1]$ and for any $s \in \mathcal{S}$,

$$\text{case } i : v_1(s) = \max_{a \in \mathcal{A}_0 \cup \{\pi_0(s)\}} \left\{ r(s, a) + \gamma \sum_{s'} p(s'|s, a)0 \right\} \tag{29}$$

$$\geq r(s, \pi_0(s)) + \gamma \sum_{s'} p(s'|s, \pi_0(s))0 \tag{30}$$

$$= L^{v_0}(s, \pi_0(s)) \geq 0 = v_0(s) \tag{31}$$

$$\text{case } ii : v_1(s) = \max_{a \in \mathcal{A}_0 \cup \{\pi_0(s)\}} \left\{ r(s, a) + \gamma \sum_{s'} p(s'|s, a)(-c) \right\} \tag{32}$$

$$\geq r(s, \pi_0(s)) + \gamma \sum_{s'} p(s'|s, \pi_0(s)))(-c) = L^{v_0}(s, \pi_0(s)) \tag{33}$$

$$= -\gamma c + r(s, \pi_0(s)) \geq -c = v_0(s) \tag{34}$$

$$\text{case } iii : v_1(s) = \max_{a \in \mathcal{A}_0 \cup \{\pi_0(s)\}} \left\{ r(s, a) + \gamma \sum_{s'} p(s'|s, a)v_0(s') \right\} \tag{35}$$

$$\geq r(s, \pi_0(s)) + \gamma \sum_{s'} p(s'|s, \pi_0(s))v_0(s') = L^{v_0}(s, \pi_0(s)) \tag{36}$$

$$\geq v_0(s) \quad \text{by assumption.} \tag{37}$$

Thus, $v_1(s_0) \geq v_1(s_0)$. For all other states $s \neq s_0$, $v_0(s) = v_1(s)$. Therefore, $v_1 \geq v_0$. Now, assume $v_n \geq \cdots \geq v_0$ with $n \geq 1$, then for any $s \in \mathcal{S}$,

$$v_{n+1}(s) = T_n v_n(s) \tag{38}$$

$$= \begin{cases} \max_{a \in \mathcal{A}_n \cup \{\pi_n(s)\}} L^{v_n}(s, a) & \text{if } s = s_n \\ v_n(s) & \text{otherwise} \end{cases} \tag{39}$$

$$\geq \begin{cases} L^{v_n}(s, \pi_n(s)) & \text{if } s = s_n \\ v_n(s) & \text{otherwise} \end{cases} \tag{40}$$

$$\geq \begin{cases} L^{v_{n-1}}(s, \pi_n(s)) & \text{if } s = s_n \quad \text{by assumption } v_n \geq v_{n-1} \\ v_{n-1}(s) & \text{otherwise .} \end{cases} \tag{41}$$

If $s_n = s_{n-1}$, then (41) is $T_{\pi_n, s_{n-1}} v_{n-1}$. By Definition 4, $T_{\pi_n, s_{n-1}} v_{n-1} = v_n$. Hence, $v_{n+1} \geq v_n$. However, if $s_n \neq s_{n-1}$, we have to do more work. There are two possible cases. The first case is that $s_n$ has been sampled for update before. That is, let $1 < j \leq n$ s.t. $s_{n-j}$ is the last time that $s_n$ is sampled for update. Then $s_n = s_{n-j}$, and $v_n(s_n) = v_{n-j+1}(s_n)$ and $\pi_n(s_n) = \pi_{n-j+1}(s_n)$. By assumption, $v_n \geq ... \geq v_{n-j} \geq ... \geq v_0$, then

$$v_{n+1}(s_n) = \max_{a \in \mathcal{A}_n \cup \{\pi_n(s_n)\}} L^{v_n}(s_n, a) \geq L^{v_n}(s_n, \pi_n(s_n)) \tag{42}$$

$$\geq L^{v_{n-j}}(s_n, \pi_n(s_n)) \quad \text{by assumption } v_n \geq v_{n-j} \tag{43}$$

$$= L^{v_{n-j}}(s_n, \pi_{n-j+1}(s_n)) \tag{44}$$

$$= T_{\pi_{n-j+1}, s_{n-j}} v_{n-j}(s_n) \quad \text{by (20)} \tag{45}$$

$$= v_{n-j+1}(s_n) \quad \text{by Definition 4} \tag{46}$$

$$= v_n(s_n). \tag{47}$$

We have just showed that $v_{n+1}(s_n) \geq v_n(s_n)$, and for all other state $s \neq s_n$, $v_{n+1}(s) = v_n(s)$. For the second case, $s_n$ has not been sampled for updated before $n$, then $v_n(s_n) = v_0(s_n)$ and $\pi_n(s_n) = \pi_0(s_n)$. By assumption, $v_n \geq ... \geq v_0$, then

$$v_{n+1}(s_n) = \max_{a \in \mathcal{A}_n \cup \{\pi_n(s_n)\}} L^{v_n}(s_n, a) \geq L^{v_n}(s_n, \pi_n(s_n)) \tag{48}$$

$$\geq L^{v_0}(s_n, \pi_n(s_n)) \tag{49}$$

$$= L^{v_0}(s_n, \pi_0(s_n)) \geq v_0(s_n) \quad \text{shown in base case} \tag{50}$$

$$= v_n(s_n). \tag{51}$$

For all other state $s \neq s_n$, $v_{n+1}(s) = v_n(s)$. Altogether, $v_{n+1} \geq v_n$ for all $n \in \mathbb{N}_0$. ∎

**Corollary 1 (Computational complexity of obtaining an $\epsilon$-optimal policy)** *Fix an $\epsilon \in (0, \|v^* - v_0\|)$, and assume DAVI initializes according to (i), (ii), or (iii) of Assumption 1. Define*

$$H_{\gamma,\epsilon} \doteq \ln\left(\frac{\|v^* - v_0\|}{\epsilon}\right) / 1 - \gamma \tag{52}$$

*as a horizon term. Then, DAVI runs for at least*

$$\tau = H_{\gamma,\epsilon}\left(\ln\left(\frac{SH_{\gamma,\epsilon}}{\delta}\right) / \ln\left(\frac{1}{1-q_{min}}\right)\right) \tag{53}$$

*iterations, returns an $\epsilon$-optimal policy $\pi_n : v_{\pi_n} \geq v^* - \epsilon\mathbb{1}$ with probability at least $1 - \delta$ using $O(mS\tau)$ elementary arithmetic and logical operations. Note that $\|v^* - v_0\|$ is unknown but it can be upper bounded by $\frac{1}{1-\gamma} + \|v_0\|$ given rewards are in $[0,1]$.*

Proof: Recall from Lemma 1, DAVI's value iterates, $v_n \to v^*$ monotonically from below (i.e., $v_n \geq v_{n-1} \geq \cdots \geq v_0$). Using this result, one can show $L^{v_n}(s, \pi_n(s)) \geq v_n(s)$ for all $s \in \mathcal{S}$ and $n \in \mathbb{N}_0$ following an induction process. We have already shown in the proof Lemma 1 that $L^{v_0}(s, \pi_0(s)) \geq v_0(s)$ for any $s \in \mathcal{S}$ in the base case. Assume that $L^{v_n}(s, \pi_n(s)) \geq v_n(s)$ for any $s \in \mathcal{S}$, we will show that $L^{v_{n+1}}(s, \pi_{n+1}(s)) \geq v_{n+1}(s)$. For any $n \in \mathbb{N}_0$ and $s_n \in \mathcal{S}$, let $\pi_{n+1}(s_n) = \arg\max_{a \in \mathcal{A}_n \cup \{\pi_n(s_n)\}} L^{v_n}(s_n, a)$ with $\pi_{n+1}(\bar{s}) = \pi_n(\bar{s})$ for all other $\bar{s} \neq s_n$.

For the case when $s = s_n$,

$$v_{n+1}(s) = T_{\pi_{n+1}, s_n} v_n(s) = L^{v_n}(s, \pi_{n+1}(s)) \tag{54}$$
$$\leq L^{v_{n+1}}(s, \pi_{n+1}(s)) \quad \text{by } v_n \leq v_{n+1}. \tag{55}$$

For the case when $s \neq s_n$, then $v_{n+1}(s) = v_n(s)$ and $\pi_{n+1}(s) = \pi_n(s)$, and thus

$$v_{n+1}(s) = T_{\pi_{n+1}, s_n} v_n(s) = v_n(s) \leq L^{v_n}(s, \pi_n(s)) \quad \text{by assumption} \tag{56}$$
$$= L^{v_{n+1}}(s, \pi_{n+1}(s)). \tag{57}$$

Altogether, we get $L^{v_{n+1}}(s, \pi_{n+1}(s)) \geq v_{n+1}(s)$ for any $s \in \mathcal{S}$, which concludes the induction.

Now, we show that $v_{\pi_n} \geq v_n$ for any $n \in \mathbb{N}_0$ using the result $L^{v_n}(s, \pi_n(s)) \geq v_n(s)$ for any $s \in \mathcal{S}$ and $n \in \mathbb{N}_0$. Fix $n$ and if we are to apply the policy evaluation operator $T_{\pi_n}$ that satisfy Lemma 3(1) to every state $s \in \mathcal{S}$, then we obtain

$$T_{\pi_n} v_n(s) = L^{v_n}(s, \pi_n(s)) \geq v_n(s). \tag{58}$$

Therefore, $T_{\pi_n} v_n \geq v_n$. By applying the $T_{\pi_n}$ operator to $T_{\pi_n} v_n \geq v_n$ repeatedly and by using the monotonicity of $T_{\pi_n}$, we have for any $k \geq 0$,

$$T_{\pi_n}^k v_n \geq T_{\pi_n}^{k-1} v_n \geq \cdots \geq v_n. \tag{59}$$

By taking limits of both sides of $T_{\pi_n}^k v_n \geq v_n$ as $k \to \infty$, we get $v_{\pi_n} \geq v_n$. Therefore,

$$0 \leq v^* - v_{\pi_n} \leq v^* - v_n \implies \|v^* - v_{\pi_n}\| \leq \|v^* - v_n\|. \tag{60}$$

Next, recall from the proof of Theorem 2 that for a given $l \in \mathbb{N}$, and with probability $1 - \delta$, $v_n$ of DAVI would have $\gamma$-contracted at least $l$ times: $\|v^* - v_n\| \leq \gamma^l \|v^* - v_0\|$, with $n \geq l \lceil \ln\left(\frac{Sl}{\delta}\right) / \ln\left(\frac{1}{1-q_{min}}\right) \rceil$. Following from (60), with probability $1 - \delta$,

$$\|v^* - v_{\pi_n}\| \leq \|v^* - v_n\| \leq \gamma^l \|v^* - v_0\|. \tag{61}$$

By setting $\gamma^l \|v^* - v_0\| = \epsilon$ and solve for $l$, we get:

$$l = \ln\frac{\|v^* - v_0\|}{\epsilon} / \ln\left(\frac{1}{\gamma}\right). \tag{62}$$

We observe that $\ln\left(\frac{\|v^* - v_0\|}{\epsilon}\right) / \ln\left(\frac{1}{\gamma}\right) \leq \ln\left(\frac{\|v^* - v_0\|}{\epsilon}\right) / (1 - \gamma) \doteq H_{\gamma,\epsilon}$. To compute $v_n$, DAVI takes $O(mS)$ elementary arithmetic operations. With probability $1 - \delta$, DAVI obtains an $\epsilon$-optimal policy with

$$O(mSn) = O\left(mSH_{\gamma,\epsilon} \ln\left(\frac{SH_{\gamma,\epsilon}}{\delta}\right) / \ln\left(\frac{1}{1-q_{min}}\right)\right) \tag{63}$$

arithmetic and logical operations. ∎

**Corollary 2 (Computational complexity of obtaining an optimal policy)** *Assume DAVI initializes according to (i), (ii), or (iii) of Assumption 1. Define the horizon term*

$$H_{\gamma, \Delta^{v^*}} \doteq \ln\left(\frac{\|v^* - v_0\|}{\Delta^{v^*}}\right) / (1 - \gamma),$$ (64)

*where $\Delta^{v^*}$ is the optimality capture region defined in Definition 5. Then, DAVI returns an optimal policy $\pi^* \in \Pi^*$ with probability $1 - \delta$, requiring*

$$O\left(mSH_{\gamma, \Delta^{v^*}} \ln\left(\frac{SH_{\gamma, \Delta^{v^*}}}{\delta}\right) / \ln\left(\frac{1}{1 - q_{min}}\right)\right)$$ (65)

*elementary arithmetic operations. Note that $\|v^* - v_0\|$ is unknown but it can be upper bounded by $\frac{1}{1-\gamma} + \|v_0\|$ given rewards are in $[0, 1]$.*

Proof: We first show that any $\pi_n$ such that $v^{\pi_n} > v^* - \Delta^{v^*}\mathbb{1}$ is an optimal policy. We prove this by contradiction. Assume $\pi_n$ is not optimal but satisfies $v_{\pi_n} > v^* - \Delta^{v^*}\mathbb{1}$, then for any $s \in \mathcal{S}$

$$L^{v^*}(s, \pi_n(s)) < L^{v^*}(s, \pi^*(s))$$ (66)

$$\implies L^{v^*}(s, \pi^*(s)) - L^{v^*}(s, \pi_n(s)) > 0$$ (67)

$$\implies L^{v^*}(s, \pi^*(s)) - L^{v^*}(s, \pi_n(s)) \geq \Delta^{v^*} \quad \text{by Definition 5}$$ (68)

$$\implies L^{v^*}(s, \pi^*(s)) - L^{v_{\pi_n}}(s, \pi_n(s)) \geq \Delta^{v^*}$$ (69)

$$\implies v^*(s) - v_{\pi_n}(s) \geq \Delta^{v^*}$$ (70)

$$\implies v_{\pi_n}(s) \leq v^*(s) - \Delta^{v^*}.$$ (71)

This contradicts the assumption and $\pi_n$ must be optimal. It is straight-forward to show that the result of Corollary 1 still holds if we require $\pi_n : v_{\pi_n} > v^* - \epsilon\mathbb{1}$ instead of $\pi_n : v_{\pi_n} \geq v^* - \epsilon\mathbb{1}$. We can then apply this result to show that DAVI returns policy $\pi_n$ such that $\pi_n : v_{\pi_n} > v^* - \Delta^{v^*}\mathbb{1}$, and thus an optimal policy, with probability $1 - \delta$ within

$$O\left(mSH_{\gamma, \Delta^{v^*}}\left(\ln\left(\frac{SH_{\gamma, \Delta^{v^*}}}{\delta}\right) / \ln\left(\frac{1}{1 - q_{min}}\right)\right)\right)$$ (72)

arithmetic and logical operations. ∎

Now we show an alternative proof to the convergence of DAVI with any initialization. Before we prove the main result, we define the following supporting lemmas.

**Lemma 5 (Williams and Baird III (1993))** *Let $v, u \in \mathbb{R}^S, s \in \mathcal{S}$.*

*Let $\pi(s) = \arg\max_{a \in \mathcal{A}} L^v(s, a)$ and an $a \in \mathcal{A}$ satisfies $L^u(s, a) \geq L^u(s, \pi(s))$. Then*

$$\|v - u\| < \frac{\Delta^v}{2\gamma}$$ (73)

*implies that $L^v(s, \pi(s)) = L^v(s, a)$.*

**Lemma 6** *Given $v \in \mathbb{R}^S$ which satisfies $\|v^* - v\| < \frac{\Delta^{v^*}}{2\gamma}$ (i.e., $v$ is inside the optimality capture region), if an action $a$ satisfies $L^v(s, a) = \max_{a' \in \mathcal{A}} L^v(s, a')$, then $a$ is an optimal action at $s$.*

Proof: For any $s \in \mathcal{S}$, let the optimal policy at $s$ be $\pi^*(s) = \arg\max_{a \in \mathcal{A}} L^{v*}(s, a)$ and $\pi(s) = \arg\max_{a \in \mathcal{A}} L^v(s, a)$, then

$$L^v(s, \pi(s)) \geq L^v(s, \pi^*(s)).$$ (74)

Since $\|v^* - v\| < \frac{\Delta^{v^*}}{2\gamma}$ and by Lemma 5, $L^{v^*}(s, \pi^*(s)) = L^{v^*}(s, \pi(s))$. ∎

**Lemma 7 (Stochastically always (Williams and Baird III, 1993))** *Let $X$ be a set of finite operators on $\mathcal{A}^S \times \mathbb{R}^S$. We say a stochastic process is stochastic always if every operator in $X$ has a non-zero probability of being drawn. Let $\Sigma$ be an infinite sequence operator from $X$ generated by a stochastic always stochastic process. Let $\Sigma'$ be a given finite sequence of operators from $X$, then*

1. $\Sigma'$ appears as a contiguous subsequence of $\Sigma$ with probability 1, and

2. $\Sigma'$ appears infinitely often as a contiguous subsequence of $\Sigma$ with probability 1.

**Theorem 3 (Convergence of DAVI with any initialisation)** *Let $\tilde{\mathcal{A}}$ be some arbitrary action subset of $\mathcal{A}$, and let $X = \{I_{\tilde{\mathcal{A}},s}, T_s | s \in \mathcal{S}\}$ be a set of DAVI operators that operate on $\mathcal{A}^S \times \mathbb{R}^S$ that is the joint space of policy and value function, where*

$$\pi_{n+1}(s) = I_{\tilde{\mathcal{A}}, s_n} \pi_n(s) = \begin{cases} \arg\max_{a \in \tilde{\mathcal{A}} \cup \{\pi_n(s)\}} L^{v_n}(s, a) & \text{if } s = s_n \\ \pi_n(s) & \text{otherwise,} \end{cases} \quad (75)$$

*and*

$$v_{n+1}(s) = T_{s_n} v_n(s) = \begin{cases} L^{v_n}(s, \pi_{n+1}(s)) & \text{if } s = s_n \\ v_n(s) & \text{otherwise.} \end{cases} \quad (76)$$

*Recall $\Pi$ is a set of deterministic policies defined in Section 2 and $\pi^* \in \Pi$. Without loss of generality, we write $\mathcal{S} = 1, ..., S$. If DAVI performs the following sequence of operations in some fixed order,*

$$I_{\tilde{\mathcal{A}}_1,1} T_1 I_{\tilde{\mathcal{A}}_2,2} T_2 ... I_{\tilde{\mathcal{A}}_S,S} T_S, \quad (77)$$

*where $\tilde{\mathcal{A}}_i$ contains the optimal action $\pi^*(i)$ for state $i$, then $v_n$ would have $\gamma$-contracted at least once by the same argument as in the proof of Theorem 2. Let $\Sigma'$ be a concatenation of $l$ copies of a sequence (77). Then, after having performed all the operations in $\Sigma'$, $v_n$ would have $\gamma$-contracted $l$ times. If $l$ satisfies:*

$$\gamma^l \|v^* - v_n\| < \frac{\Delta^{v^*}}{2\gamma}, \quad (78)$$

*then $v_n$ is inside the optimality capture region defined in Definition 5. Once inside the optimality capture region, by Lemma 6, all policies $\pi_n$ are optimal thereafer. We know from Lemma 3 (1), $\lim_{n \to \infty} T_{\pi^*} v = v^*$ and by Lemma 2 (Boundedness), all $v_n$'s are bounded. Then, the convergence of DAVI with any initialization is ensured as long as all of the states are sampled for update infinitely often.*

*The only question is whether if $\Sigma'$ would ever exist in an infinite sequence $\Sigma$ that is generated by running DAVI forever. To show that such event happens with probability 1, we apply Lemma 7. To apply Lemma 7 (Stochastically always), $X$ must be finite, which indeed it is since the state and action space are finite. Ensuring that the $\tilde{q}(s, a) > 0$ guarantees every operator in $X$ is drawn with a non-zero probability. Therefore, the stochastic process generated by running DAVI would satisfy all the properties of Lemma 7. By Lemma 7, running DAVI forever will generate any contiguous subsequence $\Sigma'$ infinitely often with probability 1.*

## B  Theoretical analysis of Asynchronous VI

Bertsekas and Tsitsiklis (1996) and Williams and Baird III (1993) have shown Asynchronous VI converges. We can view Asynchronous VI as a special case of DAVI if the subset of actions sampled in each iteration is the entire action space. That is for any $s \in \mathcal{S}$, $v \in \mathbb{R}^S$ and $\pi \in \Pi$, $\max_{a \in \mathcal{A} \cup \{\pi(s)\}} L^v(s, a) = \max_{a \in \mathcal{A}} L^v(s, a)$. We can follow similar reasoning to the proof of the convergence rate of DAVI (Theorem 2 )and show the convergence rate of Asynchronous VI with the $T$ operator defined in Definition 6. However, the sequence of increasing indices $(n_k)_{k=0}^{\infty}$, where $n_0 = 0$ in Theorem 2 takes on a slightly different meaning. In particular, between the $n_k$-th and $n_{k+1}$-th iteration, all $s \in \mathcal{S}$ have been updated at least once. Finally, the computational complexity bound of Asynchronous VI is similar to the computational complexity bound of DAVI with $p_{min} = \min_s p(s)$ instead of $q_{min}$. The computational complexity result is proven similarly to the proof of Corollary 1 found in Appendix A.

**Definition 6 (Asynchronous VI operator)** *Recall $T_{s_n} : \mathbb{R}^S \to \mathbb{R}^S$. For a given $s_n \in \mathcal{S}$, and for all $s \in \mathcal{S}$ and $v \in \mathbb{R}^S$,*

$$T_{s_n} v(s) \doteq \begin{cases} \max_{a \in \mathcal{A}} L^v(s, a) & \text{if } s = s_n \\ v(s) & \text{otherwise.} \end{cases} \quad (79)$$

*Then the iterates of Asynchronous VI evolves according to $v_{n+1} = T_{s_n} v_n$ for all $n \in \mathbb{N}_0$.*

**Lemma 8 (Asynchronous VI Monotonicity)** *The iterates of Asynchronous VI, $(v_n)_{n\geq 0}$ is a monotonically increasing sequence: $v_n \leq v_{n+1}$ for all $n \in \mathbb{N}_0$, if $r(s,a) \in [0,1]$ for any $s, a \in \mathcal{S} \times \mathcal{A}$ and if Asynchronous VI is initialized according to (i) or (ii) of Assumption 1.*

Proof: We show $(v_n)_{n\geq 0}$ is a monotonically increasing sequence by induction. All inequalities between vectors henceforth are element-wise. Let $(s_0, s_1, ..., s_n, s_{n+1})$ be the sequence of states sampled for update from iteration 1 to $n + 1$. By straight-forward calculation, we show $v_1 \geq v_0$. For all rewards in $[0, 1]$ and any $s \in \mathcal{S}$,

$$\text{case } i : v_1(s) = \max_{a \in \mathcal{A}} \left\{ r(s,a) + \gamma \sum_{s'} p(s'|s,a)0 \right\} \geq v_0(s) \tag{80}$$

$$\text{case } ii : v_1(s) = \max_{a \in \mathcal{A}} \left\{ r(s,a) + \gamma \sum_{s'} p(s'|s,a)(-c) \right\} \tag{81}$$

$$= \max_{a \in \mathcal{A}} \left\{ -\gamma c + r(s,a) \right\} \geq v_0(s). \tag{82}$$

Thus, $v_1(s_0) \geq v_0(s_0)$. For all other states $s \neq s_0, v_0(s) = v_1(s)$. Therefore, $v_1 \geq v_0$. Now, assume $v_n \geq \cdots \geq v_0$ with $n \geq 1$, then for any $s \in \mathcal{S}$,

$$v_{n+1}(s) = T_{s_n} v_n(s) \tag{83}$$

$$= \begin{cases} \max_{a \in \mathcal{A}} L^{v_n}(s,a) & \text{if } s = s_n \\ v_n(s) & \text{otherwise} \end{cases} \tag{84}$$

$$\geq \begin{cases} \max_{a \in \mathcal{A}} L^{v_{n-1}}(s,a) & \text{if } s = s_n \quad \text{by assumption } v_n \geq v_{n-1} \\ v_{n-1}(s) & \text{otherwise .} \end{cases} \tag{85}$$

If $s_n = s_{n-1}$, then (85) is $T_{s_{n-1}} v_{n-1}$. By Definition 6, $T_{s_{n-1}} v_{n-1} = v_n$. Hence, $v_{n+1} \geq v_n$. However, if $s_n \neq s_{n-1}$, we have to do more work. There are two possible cases. The first case is that $s_n$ has been sampled before. That is, let $1 < j \leq n$ s.t. $s_{n-j}$ is the last time that $s_n$ is sampled for update. Then $s_n = s_{n-j}$, and $v_n(s_n) = v_{n-j+1}(s_n)$. By assumption, $v_n \geq ... \geq v_{n-j} \geq ... \geq v_0$, then

$$v_{n+1}(s_n) = \max_{a \in \mathcal{A}} L^{v_n}(s_n, a) \tag{86}$$

$$\geq \max_{a \in \mathcal{A}} L^{v_{n-j}}(s_{n-j}, a) \quad \text{by assumption } v_n \geq v_{n-j} \tag{87}$$

$$= T_{s_{n-j}} v_{n-j}(s_{n-j}) = v_{n-j+1}(s_{n-j}) = v_n(s_n). \tag{88}$$

We have just showed that $v_{n+1}(s_n) \geq v_n(s_n)$, and for all other state $s \neq s_n, v_{n+1}(s) = v_n(s)$. For the second case, $s_n$ has not been sampled before $n$, then $v_n(s_n) = v_0(s_n)$. By assumption, $v_n \geq ... \geq v_0$, then

$$v_{n+1}(s_n) = \max_{a \in \mathcal{A}} L^{v_n}(s_n, a) \tag{89}$$

$$\geq \max_{a \in \mathcal{A}} L^{v_0}(s_n, a) \quad \text{by assumption } v_n \geq v_0 \tag{90}$$

$$\geq v_0(s_n) \quad \text{shown in base case.} \tag{91}$$

For all other state $s \neq s_n, v_{n+1}(s) = v_n(s)$. Altogether, $v_{n+1} \geq v_n$ for all $n \in \mathbb{N}_0$. ∎

**Theorem 4 (Convergence rate of Asynchronous VI)** *Assume $p(s) > 0$ and $r(s,a) \in [0,1]$ for any $s, a \in \mathcal{S} \times \mathcal{A}$, and also assume Asynchronous VI initialises according to (i), (ii) of Assumption 1. With $\gamma \in [0, 1)$ and probability $1 - \delta$, the iterates of Asynchronous VI, $(v_n)_{n\geq 0}$ converges to $v^*$ at a near-geometric rate. In particular, with probability $1 - \delta$, for a given $l \in \mathbb{N}$,*

$$\|v^* - v_n\| \leq \gamma^l \|v^* - v_0\|, \tag{92}$$

*for any n satisfying*

$$n \geq l \left\lceil \ln\left(\frac{Sl}{\delta}\right) \bigg/ \ln\left(\frac{1}{1 - p_{min}}\right) \right\rceil, \tag{93}$$

*where $p_{min} = \min_s p(s)$.*

Proof: Recall from Lemma 8, we have shown the iterates of Asynchronous VI, $v_n \to v^*$ monotonically from below. We define $(n_k)_{k=0}^{\infty}$ to be a sequence of increasing indices, where $n_0 = 0$, such that between the $n_k$-th and $n_{k+1}$-th iteration, all state $s \in \mathcal{S}$ have been updated at least once. At the $n_{k+1}$-th iteration, $v_{n_{k+1}} \geq T_{\pi^*} v_{n_k}$. This implies that at the $n_{k+1}$-th iteration, Asynchronous VI would have $\gamma$-contracted at least once:

$$0 \leq v^* - v_{n_{k+1}} \leq v^* - T_{\pi^*} v_{n_k}, \implies \|v^* - v_{n_{k+1}}\| \leq \|v^* - T_{\pi^*} v_{n_k}\|, \tag{94}$$

$$\|v^* - T_{\pi^*} v_{n_k}\| = \|T_{\pi^*} v^* - T_{\pi^*} v_{n_k})\| \leq \gamma \|v^* - v_{n_k}\| \tag{95}$$

$$\implies \|v^* - v_{n_{k+1}}\| \leq \gamma \|v^* - v_{n_k}\|. \tag{96}$$

The probability of the failure event

$$\mathbb{P}(\mathcal{E}^c) \leq \sum_{i=1}^{l} \sum_{s \in \mathcal{S}} \mathbb{P}(\mathcal{E}_i^c) \tag{97}$$

$$\leq Sl(1 - p_{min})^N \tag{98}$$

with $p_{min} = \min_{s \in \mathcal{S}} p(s)$ instead of $q_{min}$. The rest follows similar reasoning to the proof of Theorem 2 and obtain the result. ∎

**Corollary 3 (Computational complexity of Asynchronous VI)** *Fix an $\epsilon \in (0, \|v^* - v_0\|)$, and assume Asynchronous VI initialises according to (i) or (ii) of Assumption 1. Define*

$$H_{\gamma,\epsilon} \doteq \ln\left(\frac{\|v^* - v_0\|}{\epsilon}\right) / 1 - \gamma \tag{99}$$

*as a horizon term. Then, Asynchronous VI returns an $\epsilon$-optimal policy $\pi_n : v_{\pi_n} \geq v^* - \epsilon\mathbb{1}$ with probability at least $1 - \delta$ using*

$$O\left(ASH_{\gamma,\epsilon}\left(\ln\left(\frac{SH_{\gamma,\epsilon}}{\delta}\right) / \ln\left(\frac{1}{1 - p_{min}}\right)\right)\right) \tag{100}$$

*elementary arithmetic and logical operations. Note that $\|v^* - v_0\|$ is unknown but it can be upper bounded by $\frac{1}{1-\gamma} + \|v_0\|$ given rewards are in $[0, 1]$.*

Proof: Recall from Lemma 8, the iterates of Asynchronous VI, $v_n \to v^*$ monotonically from below (i.e., $v_n \geq v_{n-1} \geq \cdots \geq v_0$). For any $n \in \mathbb{N}_0$ and $s_n \in \mathcal{S}$, let $\pi_{n+1}(s_n) = \arg\max_{a \in \mathcal{A}} L^{v_n}(s_n, a)$ with $\pi_{n+1}(\bar{s}) = \pi_n(\bar{s})$ for all other $\bar{s} \neq s_n$. One can show $L^{v_n}(s, \pi_n(s)) \geq v_n(s)$ for any $s \in \mathcal{S}$ and $n \in \mathbb{N}_0$ following a similar argument as in the proof of Corollary 1. Now, we show $v_{\pi_n} \geq v_n$ for any $n \in \mathbb{N}_0$. Fix $n$ and if we are to apply the policy evaluation operator $T_{\pi_n}$ that satisfy Lemma 3(1) to every state $s \in \mathcal{S}$, then we obtain

$$T_{\pi_n} v_n(s) = L^{v_n}(s, \pi_n(s)) \geq v_n(s). \tag{101}$$

Therefore, $T_{\pi_n} v_n \geq v_n$. By applying the $T_{\pi_n}$ operator to $T_{\pi_n} v_n \geq v_n$ repeatedly, and by using the monotonicity of $T_{\pi_n}$, we have for any $k \geq 0$,

$$T_{\pi_n}^k v_n \geq T_{\pi_n}^{k-1} v_n \geq \cdots \geq v_n. \tag{102}$$

By taking limits of both sides of $T_{\pi_n}^k v_n \geq v_n$ as $k \to \infty$, we get $v_{\pi_n} \geq v_n$. Therefore,

$$0 \leq v^* - v_{\pi_n} \leq v^* - v_n \implies \|v^* - v_{\pi_n}\| \leq \|v^* - v_n\|. \tag{103}$$

Next, recall from the proof of Theorem 4 that for a given $l \in \mathbb{N}$, and with probability $1 - \delta$, $v_n$ of Asynchronous VI would have $\gamma$-contracted at least $l$ times (i.e., $\|v^* - v_n\| \leq \gamma^l \|v^* - v_0\|$) with $n \geq l\left\lceil \ln\left(\frac{Sl}{\delta}\right) / \ln\left(\frac{1}{1-p_{min}}\right) \right\rceil$. Following from (103), with probability $1 - \delta$,

$$\|v^* - v_{\pi_n}\| \leq \|v^* - v_n\| \leq \gamma^l \|v^* - v_0\|. \tag{104}$$

By setting $\gamma^l \|v^* - v_0\| = \epsilon$ and solve for $l$, we get:

$$l = \ln\frac{\|v^* - v_0\|}{\epsilon} / \ln\left(\frac{1}{\gamma}\right). \tag{105}$$

We observe that $\ln\left(\frac{\|v^* - v_0\|}{\epsilon}\right)/\ln\left(\frac{1}{\gamma}\right) \leq \ln\left(\frac{\|v^* - v_0\|}{\epsilon}\right)/(1 - \gamma) \doteq H_{\gamma,\epsilon}$. To compute $v_n$, Asynchronous VI takes $O(AS)$ elementary arithmetic operations. With probability $1 - \delta$, Asynchronous VI obtains an $\epsilon$-optimal policy within

$$O(ASn) = O\left(ASH_{\gamma,\epsilon}\left(\ln\left(\frac{SH_{\gamma,\epsilon}}{\delta}\right)\Big/\ln\left(\frac{1}{1 - p_{min}}\right)\right)\right) \tag{106}$$

arithmetic and logical operations. ∎

## C   More experiments

In this section, we show additional experiments with the MDPs described in Section 6 with rewards generated via a standard normal and a Pareto distribution.

Recall that the experiments were set up to see how DAVI's performance is affected by the sparsity of rewards. Pareto distribution with a shape of 2.5 is a "heavy-tail" distribution, and the rewards sampled from this distribution could result in a few large values. On the other hand, the rewards sampled via the standard Normal distribution could result in many similar values. We hypothesize that DAVI would converge faster than Asynchronous VI in domains with multiple optimal or near-optimal policies, which could be the case in the normal-distributed reward setting.

The algorithms that will be compared in the experiments are VI, Asynchronous VI, and DAVI. We implement Asynchronous VI and DAVI using uniform sampling to obtain the states. DAVI samples a new set of actions via uniform sampling without replacement in each iteration.

### C.1   Single-state experiment

This experiment consists of a single-state MDP with 10000 actions, and all terminate immediately. We experiment with two reward distributions: Pareto-reward and Normal-reward. For Pareto-reward, all actions have rewards generated according to a Pareto distribution with shape 2.5. For Normal-reward, all actions have rewards generated according to the standard normal distribution.

### C.2   Multi-reward experiment

This experiment consists of two MDPs. The first set consists of a tree with a depth of 2. Each state has 50 actions, where each action leads to 2 other distinct next states. All actions terminate at the leaf states. In one setting, the rewards are distributed according to the Pareto distribution with a shape of 2.5. In the other setting, the rewards are distributed according to the normal distribution.

The second set of MDPs consists of a random MDP with 100 states, where each state has 1000 actions. Each action leads to 10 next states randomly selected from the 100 states with equal probability. All transitions have a 0.1 probability of terminating. In one setting, the rewards are distributed according to the Pareto distribution with a shape of 2.5. In the other setting, the rewards are distributed according to the standard normal distribution. The $\gamma$ in all of the MDPs are 1.

### C.3   Discussion

Figure 3 and Figure 4 show the performance of the algorithms. All graphs included error bars showing the standard error of the mean. All graphs smoothly increased towards the asymptote except for Asynchronous VI in Figure 3 and VI in Figure 4, whose performances were step-functions [2]. The y-axis of each graph showed a state value averaged over 200 runs. The x-axes showed run-times, which have been adjusted for computations.

In Figure 3, DAVI with $m = 1$ was significantly different from that of DAVI with $m = 10, 100, 1000$. However, in the Normal-reward setting, the performance of DAVI with $m = 1$ was much closer to the performance of DAVI with $m = 10, 100, 1000$. In the Pareto-reward setting, where there could only be a few large rewards, the results were similar to that of the needle-in-the-haystack setting of Figure 1. In the Normal-reward setting, where most of the rewards were similar and concentrated around 0, the results were similar to that of the multi-reward setting of Figure 1.

---

[2]Asynchronous VI is equivalent to VI in the single-state experiment since there is only one state.

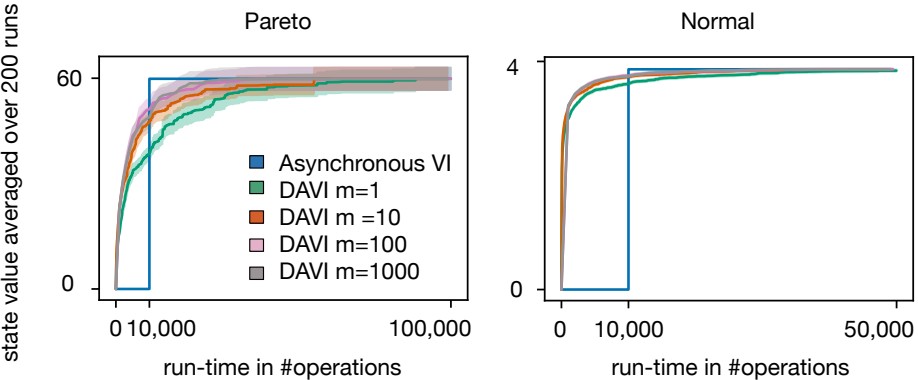

Figure 3: Single-state experiment with 10000 actions: (a) rewards are Pareto distributed with shape 2.5 (b) rewards are standard normal distributed. The Asynchronous VI in this experiment is equivalent to VI since there is only one state. We run each instance 200 times with a new MDP generated each time. In each run, each algorithm is initialized to 0.

In Figure 4 in both tree Pareto-reward and Normal-reward settings (top row), DAVI with $m = 1$ was significantly different from that of DAVI $m = 10$. In the tree setting, with normal-distributed rewards, where there may be multiple actions with similarly large rewards, DAVI $m = 10$ converged faster than VI and Asynchronous VI.

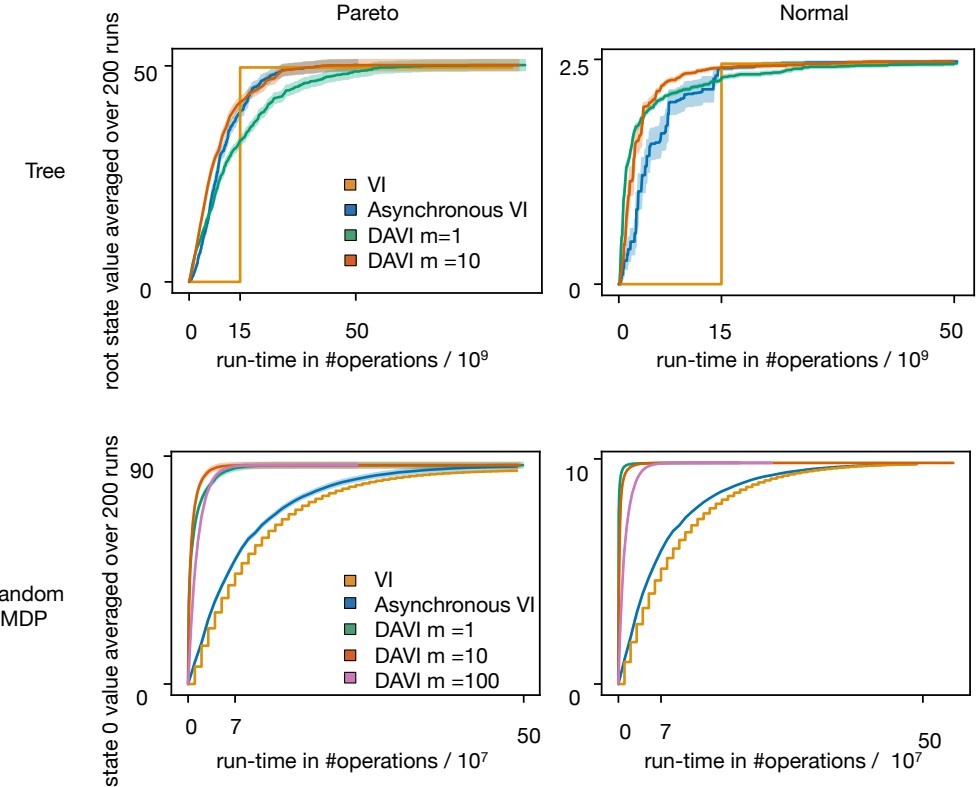

Figure 4: Multi-state experiment: (top row) MDP with a tree structure with Pareto and normal distributed rewards (bottom row) random MDP with Pareto and normal distributed rewards. We run each instance 200 times with a new MDP generated each time. In each run, all algorithms are initialized to 0.

In the random-MDP setting, DAVI, for all values of $m$, converged faster than VI and Asynchronous VI in both the Pareto-reward and Normal-reward settings, as evident in the bottom row of Figure 4. As expected, DAVI converged faster than Asynchronous VI and VI in the case of multiple near-optimal policies. Note, DAVI $m = 100$ was the slowest to converge, a case where the action subset size is large. This result makes sense as Asynchronous VI with the full action space did not converge as fast as DAVI with smaller action subsets.