# OpenReview forum: "Doubly-Asynchronous Value Iteration: Making Value Iteration Asynchronous in Actions"
_NeurIPS.cc/2022/Conference — NeurIPS 2022 Accept_

### Official Review · Reviewer_baze · 2022-07-11

**Rating:** 6
**Confidence:** 3
**Soundness:** 3 good
**Presentation:** 3 good
**Contribution:** 2 fair

**Summary:**

This paper propose a dynamic-programming method with finer granularity than predecessor. The proposed method iterate on sampled actions rather than on the entire action set as seen in traditional value iteration and asynchronous value iteration. The resulting algorithm is provided with tight bounds and convergence results, with clear line of future work.


**Questions:**

Sampling distribution (state and action) may become a prominent influencer if employed empirically, would the sampling distribution be better guided with exploration techniques based on the maturity of action values?


**Limitations:**

Limited empirical performance gain.


**Strengths And Weaknesses:**

While theoretical analysis is thorough, the empirical result could be improved in complexity. A more challenging (realistic?) environment could be employed for evaluation. In addition, the proposed method does reduce the complexity of the problem by much (as expected), the entire sampling process may still be “prohibitively expansive”, which is not ideal if paired with catastrophically forgetting networks.
The paper is well written and the theoretical foundation is solid.

---

### Official Review · Reviewer_qTcp · 2022-07-13

**Rating:** 6
**Confidence:** 4
**Soundness:** 4 excellent
**Presentation:** 4 excellent
**Contribution:** 3 good

**Summary:**

The paper introduces and analyzes DAVI. A variation of value iteration that selects both the state and the action of evaluated look ahead asynchronously. It provides proof of convergence and convergence rate analysis for the algorithm as well as its computation complexity.

**Questions:**

Does authors have any more concrete conjectures about the scenarios DAVI converges faster than VI?

**Limitations:**

The algorithm's performance is precisely bounded and compared to literature.

**Strengths And Weaknesses:**

The paper is technically very precise, thorough, and well-written. It introduces a nice extension to normal value iteration, which seems to be effective in some domains. The possibility of choosing only a subset of actions to evaluate, opens up a new direction of research regarding the action selection method. One can see this line of research as equivalent of search-control queue studies.


One strange part of the paper was the emphasis on DAVI "nearly matching" computational complexity of VI. The whole promise of DAVI is to be computationally more efficient than VI, and such guarantee is not helping with establishing that. It is a nice result to have but the writing is too positive about it in my opinion. I know the complexity is for the worst-case scenario and the speed-up probably does not always happen, which leads to my second suggestion.

A possible improvement to the paper would be some analysis of the cases DAVI introduces a speed-up compared to traditional VI. The experiments suggest that the number of viable action in each state is a factor for a speed-up. I wonder if this could be formalize in the form of $Q^*$ values.

Overall, I think this is a solid paper.

---

### Official Review · Reviewer_592T · 2022-07-18

**Rating:** 8
**Confidence:** 1
**Soundness:** 4 excellent
**Presentation:** 3 good
**Contribution:** 4 excellent

**Summary:**

This paper proposes doubly-asynchronous value iteration (DAVI) that generalizes asynchrony from states to states and actions. The intuition behind DAVI is incremental maximization, where maximizing over a few actions could improve the value estimate for a certain state. DVAI maximizes over a sampled subset of actions of arbitrary size. DAVI is proved to converge to the optimal value function with probability one at a near-geometric rate with probability $1-\delta$. DAVI returns a near-optimal policy in computation time that nearly matches a previously established bound for VI.

**Questions:**

I don't have any questions

**Limitations:**

I have not seen any limitations. This paper is mathematically sound with good intuition.

**Strengths And Weaknesses:**

Strength:
1. This paper studies one of the most widely studies dynamic programming algorithm: Value iteration (VI).
2. The proposed DAVI generalizes the idea of asynchrony from states to states and actions.
3. The computational complexity bound for DAVI nearly match a previously established bound for VI and the convergence rate is nearly geometric rate.
4. DAVIS could converge to the optimal value function despite only maximizing over a subset of actions in each update.

Weakness:
I don't work in this field.

---

### Author Response · Authors · 2022-08-02
**Author's response**

We thank all reviewers for their positive feedback and suggestions for future directions. We are gratified that all the reviewers found our work novel and significant, with clear writing, technically precise theory, and thorough analysis. Reviewer 592T pointed out that DAVI “generalizes the idea of asynchrony in value iteration from states to states and actions.”  As reviewer qTcp commented, DAVI is a “nice extension to normal value iteration.” We agree with reviewer qTcp that our work opens the door to interesting future directions in designing sampling strategies to guide the exploration of state and action space. In what follows, we respond to specific reviewer questions and comments.

To address reviewer baze's comment on the complexity of the environment, the present work primarily serves to establish the theory which sets DAVI as a valid and worthwhile approach. We performed experiments on simple environments to better understand DAVI’s behavior and validate our theoretical results. Evaluating DAVI in more complex environments is an important direction for future work.

Reviewer qTcp asked whether we have more concrete conjecture on scenarios where DAVI converges faster than VI.  Aside from what is already noted in the paper, we do not have more concrete conjectures as of yet. However, this is an interesting direction for future research.

---

### Meta-Review · Area_Chair_tBKq · 2022-08-26

**Recommendation:** Accept
**Confidence:** Certain

**Metareview:**

The paper proposes and analyzes a new asynchronous value iteration method, which extends asynchrony from states to states and actions.
The reviewers have found the paper technically sound and well executed and they agree that it deserves publication.
In preparing the final version of their paper, the authors are invited to consider the reviewers' comments.

**Award:**

No

---

### Decision · Program_Chairs · 2022-09-14

Accept